# Estimating Seismic Demands of a Single-Door Electrical Cabinet System Based on the Performance Limit-State of Concrete Shear Wall Structures

**Bu-Seog Ju [1], Hoyoung Son [1], Sangwoo Lee [1],* and Shinyoung Kwag [2],***

[1] Department of Civil Engineering, Kyung Hee University, Yongin-si 17104, Korea; bju2@khu.ac.kr (B.-S.J.); shyoung0623@khu.ac.kr (H.S.)
[2] Department of Civil & Environmental Engineering, Hanbat National University, Daejeon 34158, Korea
* Correspondence: slee83@khu.ac.kr (S.L.); skwag@hanbat.ac.kr (S.K.)

**Abstract:** The electrical cabinet systems in power plants are critical non-structural components to maintaining sustainable operation and preventing unexpected accidents during extreme events. This system consists of various electrical equipment such as relays, circuit breakers, and switches enclosed by a steel cabinet for the protection of the equipment. The cabinet systems are installed in and protected by structures so that the cabinet's behavior is totally dependent on the behavior of the structures when subjected to an earthquake. Therefore, it is essential to qualify the seismic performance of the cabinet system considering the effect of the primary structure where the electrical cabinet system is mounted. In addition, with the implementation of ASCE-43 design standards for nuclear facilities, facility design allowing nonlinear behavior has gained greater attention in nuclear power plants, and research on how the response of the cabinet varies according to allowable damage levels of structures is needed. In this study, Finite Element (FE) models of a single-door electrical cabinet and concrete shear wall structure validated through experimental data are used for a decoupled analysis to estimate the seismic demands of the electrical cabinet. Three different earthquake loadings, referred to as EQ#13, #17, and #19, used in the SMART-2013 project are selected to obtain floor responses of the concrete structure, and the loadings lead to different levels of damage (minor, moderate, and major damage, respectively) to the structure. Finally, the floor responses based on levels of the damage to the primary structure are applied to the electrical cabinet system as input loadings for the decoupled analysis. Thus, this study presents the effects of the cabinet elevation and performance limit-state for concrete shear wall structures on the response of the electrical cabinet, and it shows that while the difference in seismic demands is not significant in the minor and moderate damage states, a meaningful difference occurs in the degree of the major damage state.

**Keywords:** decoupled analysis; electrical cabinet; concrete shear walls; performance-based seismic qualification



## 1. Introduction

In power plants, non-structural components (e.g., cabinets, relays, circuit breakers, and switches) are as important as structural components (e.g., shear walls, columns, and slabs) in terms of plant operation, maintenance, and accident prevention. Since the 2016 Gyeongju earthquake in Korea, the need for seismic qualification of the electrical equipment in power plants with high-frequency earthquakes has been apparent [1]. The high-frequency earthquakes rarely cause damage to structures but can cause severe damage to the non-structural components, such as electrical equipment, pipes, and ceilings [2–4]. The pieces of electrical equipment in power plants are critical non-structural components to maintaining sustainable operation and preventing unexpected accidents. They are vulnerable to damage during earthquakes, thereby being enclosed in a steel cabinet to protect it from external impacts. The electrical cabinet system must be functionally and structurally safe

during and after earthquakes. Moreover, the cabinet's behavior is totally dependent on the primary structures in which it is installed when subjected to an earthquake. In other words, the behavior of the structures directly affects the cabinet systems, and the cabinet's responses are very sensitive to the elevation at which the cabinet is mounted. Thus, it is essential to verify the seismic performance of the cabinet system considering the effect of the structures, and the floor acceleration responses of the structures corresponding to the elevation of the cabinet must be the load for the seismic qualification of the cabinet system. In previous studies, idealized models such as frame or stick models were used to assess the seismic performance of cabinets rather than detailed 3D FE models due to the complexities of the electrical cabinet system such as bolt and welding connections [5–7]. While the simplified models effectively capture the overall behavior of the cabinet system (e.g., bending behavior), it is difficult to represent aspects of the local behavior of the cabinet (e.g., shear, torsion, and vibration). In addition, it is impossible to examine the responses at desired or interesting locations of the electrical equipment in the cabinet, thereby limiting the accuracy of calculations of the In-Cabinet Response Spectrum (ICRS) at points of interest. Existing studies using decoupled or coupled analysis to consider the effect of structures on the cabinet also use a simplified structural model and evaluate the seismic response of the electrical cabinet system without validation of the developed structural model [8–10]. While general civil structures are allowed for design considering the nonlinearity of materials [11], structures and components in nuclear power plants traditionally have been designed with the objective that they remain linear when subjected to a design-level earthquake. Recently, the American Society of Civil Engineers (ASCE) published the ASCE-43 design standard [12], allowing for design by selecting seismic loads and limit-states according to the importance of structures and components instead of the conservative linear design practice. An engineer can design structures by using the four limit-states classified from A to D. Limit-state D represents the conventional elastic design, and limit-states C, B, and A represent minor, moderate, and major nonlinearity of structures (damage), respectively. Furthermore, the US Nuclear Regulatory Commission (USNRC) is interested in using limit-state C, particularly in the design of concrete shear walls [13]. Therefore, research on how the response of the cabinet system varies according to the degree of nonlinearity of the primary structures is needed. Son et al. [14] developed a 3D single-door electrical cabinet model to more accurately calculate not only the overall behavior of the cabinet but also the local responses, and the developed model was validated through comparison with an existing shake table experiment. In addition, an FE model of a three-story concrete shear wall structure used in the SMART-2013 experiment and international benchmark project was developed to evaluate the existing concrete nonlinear model and to assess the nonlinear characteristics of the shear wall structure by Lee [15], and the model was validated through the comparison with the experimental results. In this study, the concrete shear structure model was used to evaluate the responses of each floor, and a decoupled analysis of the single-door electrical cabinet was conducted using the floor responses as the input loads. Thus, the effects of the cabinet system elevation and degree of damage to the structure on the response of the electrical cabinet system can be documented.

## 2. Experimental Test and Validation of Single-Door Electrical Cabinet

### 2.1. Shaking Table Test of Single-Door Electrical Cabinet

Experiments were conducted with the electrical cabinet system at the Seismic Simulation Test Center in Korea to identify modal characteristics (natural frequency and mode shape) and understand the seismic behavior of the single-door cabinet [16]. The dimensions of the single-door cabinet are 800 × 800 × 2350 mm (width × depth × height), as shown in Figure 1. A total of six measurement points (A8–A13) were selected, and the acceleration responses at the points were collected. Figure 2 shows the measurement points on the cabinet in the experiment. A sinusoidal signal with PGA 0.07 g and a frequency range between 1 and 50 Hz was applied to identify the dynamic characteristics of the cabinet

at the measurement points. The level of amplification of the response at the points was estimated as shown in Figure 3. It was found that global modes occurred at 16 and 24 Hz, and local modes occurred at 30.5 and 37.5 Hz.

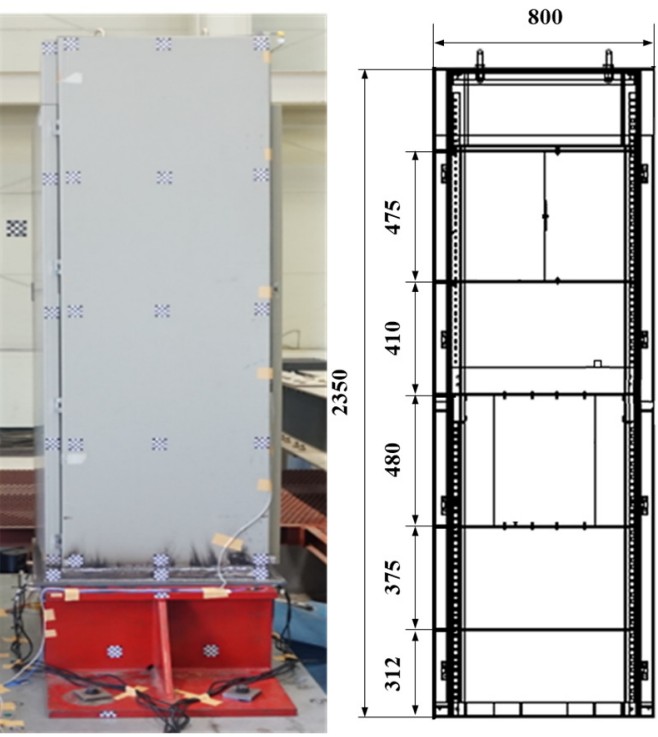

**Figure 1.** Single-door electrical cabinet [14].

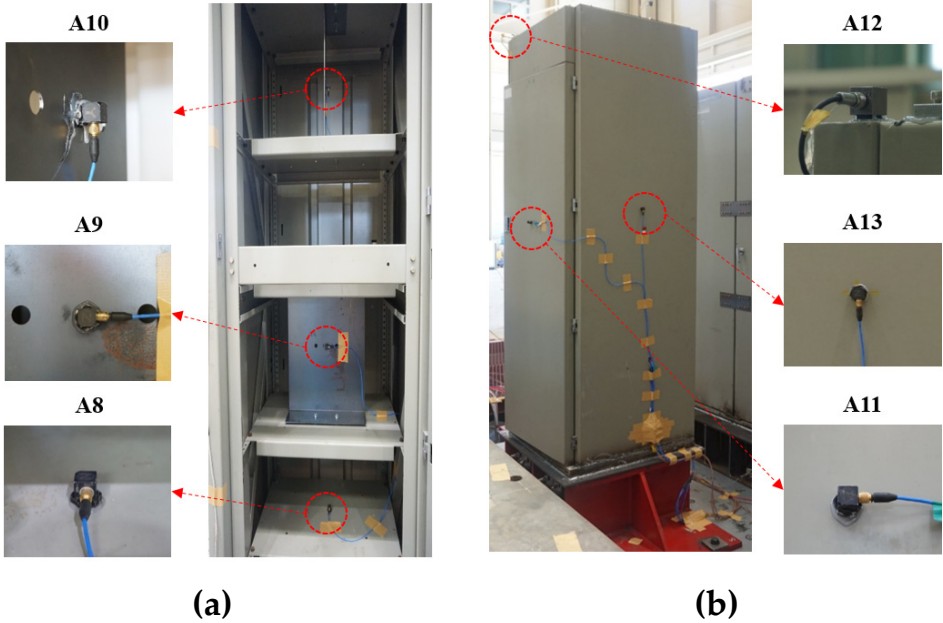

**Figure 2.** Measurement points on electrical cabinet: (**a**) internal measurement points; (**b**) external measurement points.

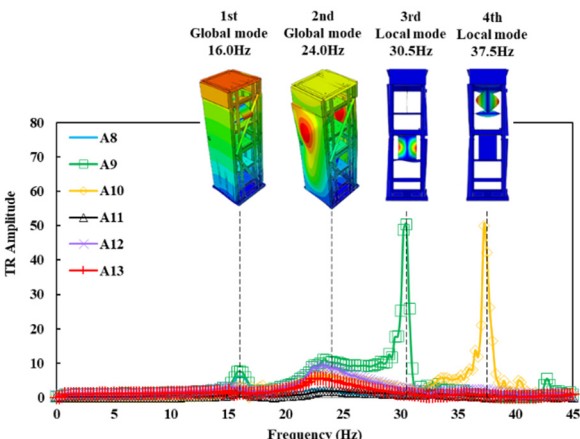

**Figure 3.** Experimental natural frequencies corresponding to numerical mode shapes [16].

### 2.2. Validation of FE Model of Electrical Cabinet

A high-fidelity 3D FE model of the electrical cabinet was developed using the commercial FE software ABAQUS [17], as shown in Figure 4. The model consists of inside and outside plates, vertical and horizontal panels, main frames, and bracings, for which a four-node shell reduced integration element was used. Elastic modulus, density, and Poisson's ratio of the cabinet were $2.1 \times 10^5$ Mpa, $7.85 \times 10^{-9}$ t/mm$^3$, and 0.28, respectively [14]. To validate the cabinet model, a modal analysis was conducted. The modal results are compared with the experimental results in Table 1. The numerical natural frequencies of the cabinet are similar to the experimental values. As observed in the experiment, the mode behaviors at the 3rd and 4th modes are localized on the second-level vertical panel (A9) and third-level vertical panel (A10), and the FE model exactly captures the local behavior in the cabinet, as presented in Figure 3. Thus, the FE model reasonably reproduces the global- and local-mode shapes, simultaneously.

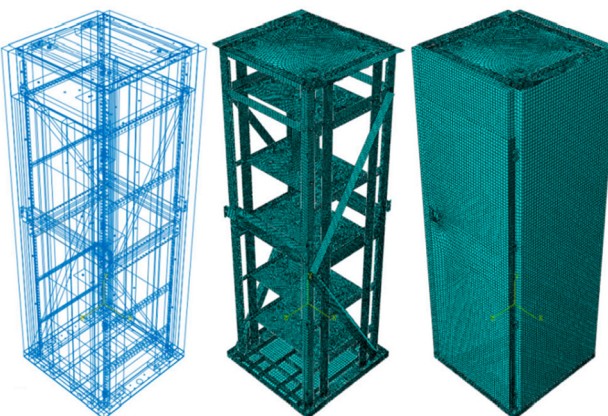

**Figure 4.** FE model of single-door electrical cabinet [14].

**Table 1.** Comparison of analytical and experimental results for natural frequency of the electrical cabinet.

| Frequency | 1st Mode (Global Mode) | 2nd Mode (Global Mode) | 3rd Mode (Local Mode) | 4th Mode (Local Mode) |
|---|---|---|---|---|
| Experiment | 16.00 Hz | 24.0 Hz | 30.5 Hz | 37.5 Hz |
| FE analysis | 16.12 Hz | 23.9 Hz | 30.6 Hz | 37.3 Hz |

## 3. Experimental Test and Validation of Concrete Shear Wall Structures

### 3.1. SMART-2013 Experiment and International Benchmark Program

As the need for accurate understanding and prediction of the nonlinear behavior of nuclear power plant structures has increased, a seismic shake table test on a 1/4 scale concrete shear wall structure and benchmark project were conducted in 2012 [18,19]. This project, referred to as the Seismic design and best-estimate Methods Assessment (SMART-2013), was organized and conducted by the French Sustainable Energies and Atomic Energy Commission (CEA) and Electricité De France (EDF). Concrete structures are commonly used in electrical facilities in nuclear power plants, and the configuration and dimensional details of the structure are shown in Figure 5. The SMART-2013 project consisted of a structural experiment and a blind prediction. In the experiment, low-level random signals in both horizontal directions (0.05 and 0.1 g) were applied to identify modal characteristics such as natural frequency and mode shape of the structure. The results were provided for benchmark participants to validate their structural models. Furthermore, the structure was subjected to a series of seismic loadings with various levels, and the structural responses were collected. The applied seismic loadings are divided into three stages. In the first stage, an artificial seismic loading of PGA 0.1 g, which is a design level load, was applied. In the second stage, the structure was successively subjected to real Northridge earthquakes between PGA 0.2 and 1.1 g to identify the nonlinear response and damage to the structure. These experimental data were used for the benchmark project. In the third stage, the Northridge aftershock loads were applied to check the response of the structure to the aftershock. A total of eight hydraulic actuators were used to apply seismic loadings in horizontal and vertical directions, respectively, and those actuators were recorded to obtain displacement and acceleration time histories during the whole experimental campaign. Moreover, both acceleration and displacement were monitored at the corners of each slab. Figure 6 shows the plan view of the SMART-2013 structure and the shaking table

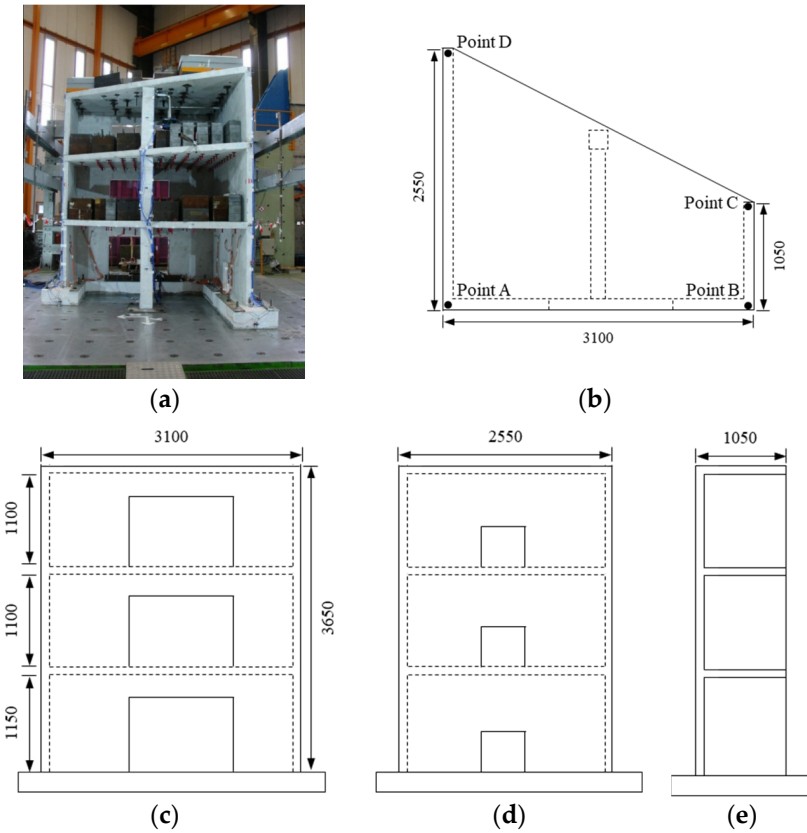

**Figure 5.** Configuration and dimensional details of the SMART-2013 structure [19,20]: (**a**) SMART-2013 structures; (**b**) floor; (**c**) shear wall #1; (**d**) shear wall #2; (**e**) shear wall #3.

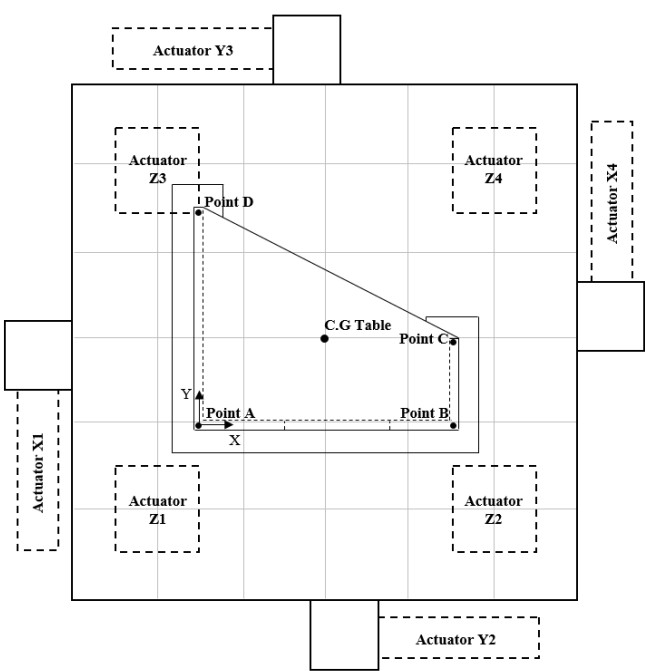

**Figure 6.** Plan view of structure and shake table in the SMART-2013 project.

Three different earthquake loadings, referred to as EQ#13, #17, and #19, used in the SMART-2013 project were selected to obtain floor responses of the concrete structure, and the loadings led to different levels of damage (minor, moderate, and major damage, respectively) to the structure. No cracks occurred in the structure during the design-level seismic loading test, and localized cracks started to occur at the bottom of shear wall #3 and the first floor during EQ#13. The propagation of the cracks was observed more and more significantly with the increase in seismic loadings, and finally, local fractures occurred at the toe of shear wall #3, resulting in an uplift of the shear wall and the detachment of rebars during EQ#19. Table 2 shows the maximum Inter-Story Drift (ISD) for EQ#13, #17, and #19 compared with the allowable ISD of concrete shear walls provided by ASCE-43 [12]. ASCE-43 is a guideline of seismic design criteria for structures, systems, and components in nuclear facilities and defines various performance limit-states for a design according to ISD. Thus, it could show that the structure suffered minor, moderate, and major levels of damage due to EQ#13, #17, and #19, respectively. Table 3 summarizes the characteristics of EQ#13, #17, and #19. In order to examine the characteristics of the seismic loadings applied, the Acceleration Response Spectrum (ARS) for EQ#13, #17, and #19 was calculated, as shown in Figure 7. Figure 7a compares the ARS with the design spectrum (0.3 g) of REG 1.60. It shows that ARS of EQ#17, and #19 envelop the design spectrum in both low (below 10 Hz) and high-frequency ranges (above 10 Hz); however, EQ#13 only envelops the design spectrum in the high-frequency range. Figure 7b compares ARS anchored at 0.273 g with REG 1.60 (0.3 g) and Uniform Hazard Response Spectrum (UHRS, 0.273 g) at Uljin in Korea, where the Hanwool nuclear power plant is located. The ground motions used in the SMART-2013 experiment contained high-frequency components, representing high-frequency motions in Korea well, thereby being appropriate for examining the effect of the high-frequency motion on the electrical cabinet based on different performance limit-states of the primary structures.

**Table 2.** Comparison of inter-story drift between the experiment and ASCE 43 [12].

| Experiment | EQ#13 | EQ#17 | EQ#19 |
|---|---|---|---|
| | 0.0039 | 0.0055 | 0.0134 |
| ASCE43 | Limit-State C (Limited permanent distortion) | Limit-State B (Moderate permanent distortion) | Limit-State A (Large permanent distortion) |
| | 0.0039 | 0.0060 | 0.0080 |

**Table 3.** Characteristics of EQ#13,17, and 19 in the SMART-2013 experiment [20].

| | PGA in x-Direction (g) | PGA in y-Direction (g) | PGA |
|---|---|---|---|
| EQ#13 | 0.40 | 0.21 | Scaled Northridge |
| EQ#17 | 0.60 | 0.40 | Scaled Northridge |
| EQ#19 | 1.10 | 1.00 | Real Northridge |

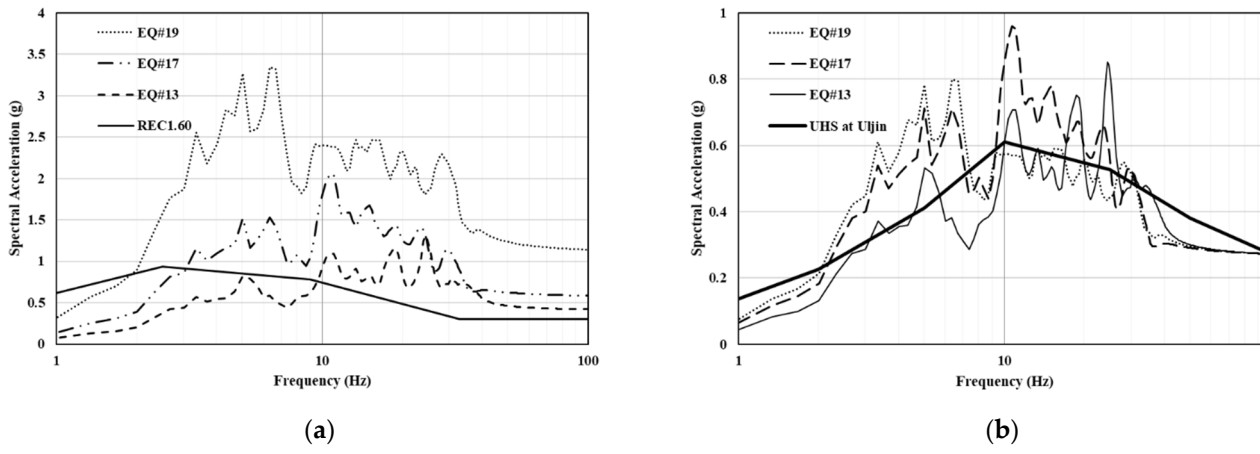

(a)                                                                                          (b)

**Figure 7.** ARS for EQ#13,17, and 19 [21,22]; (**a**) comparison with REG 1.60; (**b**) comparison with UHRS at Uljin.

### 3.2. Validation of Finite Element Model of SMART-2013 Structure

In this study, a decoupled analysis of the electrical cabinet system was performed by the FE model of SMART-2013 developed using the commercial FE software ABAQUS [17]. This structural model consists of shear walls, slabs, shake table, columns, beams, and foundations. A four-node shell reduced integration element was used for the first three parts, and an eight-node solid reduced integration element was used for the other parts. The developed model is shown in Figure 8. To validate the structural model, the modal result was compared with the experimental one. Table 4 shows the natural frequencies calculated through the modal analysis. It shows that the numerical frequencies of the structure are similar to the experimental one. In addition, the first mode is governed by horizontal behavior in the x-direction, the second mode is governed by horizontal behavior in the y-direction, and the third mode shows rotational behavior, which is the same as the experimental responses.

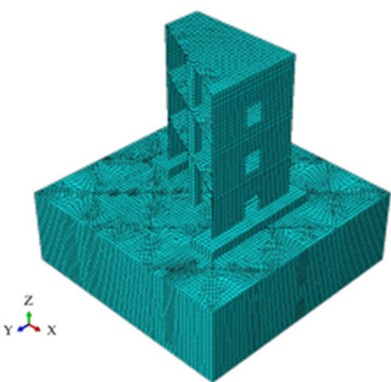

**Figure 8.** FE model of the SMART-2013 structure [15].

**Table 4.** Comparison of experimental and analytical modal frequencies of the SMART-2013 concrete structure [18].

| Frequency | 1st Mode | 2nd Mode | 3rd Mode |
|---|---|---|---|
| Experiment | 6.28 Hz | 7.86 Hz | 16.50 Hz |
| FE analysis | 6.17 Hz | 8.58 Hz | 16.12 Hz |

## 4. Building-Cabinet Interaction: Decoupled Analysis

A decoupled analysis of the single-door electrical cabinet was performed with the FE model of the concrete shear wall structure to investigate the cabinet's behavior according to the floor level where the cabinet was mounted and the degree of damage to the structure. EQ#13, #17, and #19 were applied to the SMART-2013 structural model to examine acceleration time responses on each floor (Bottom, 1st, 2nd, and 3rd floor). Based on the SMART-2013 experiment report [23] and the seismic analysis of the FE model of the SMART-2013 structure, the maximum response occurs at point D; thus, the acceleration responses in three directions at point D were collected for all floors. The collected floor acceleration time responses were applied to the electrical cabinet model as input ground motions in the three directions, as presented in Figure 9. Figure 10 shows the Floor Response Spectrum (FRS) obtained from the SMART-2013 structural model for EQ#13, #17, and #19.

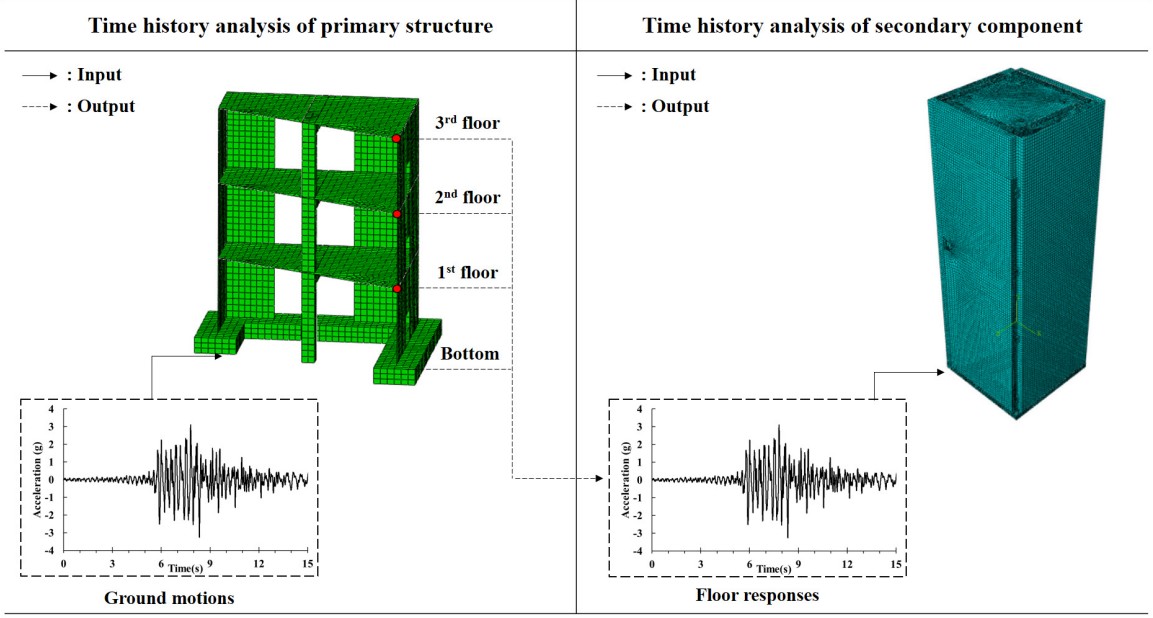

**Figure 9.** Scheme of decoupled analysis of the electrical cabinet.

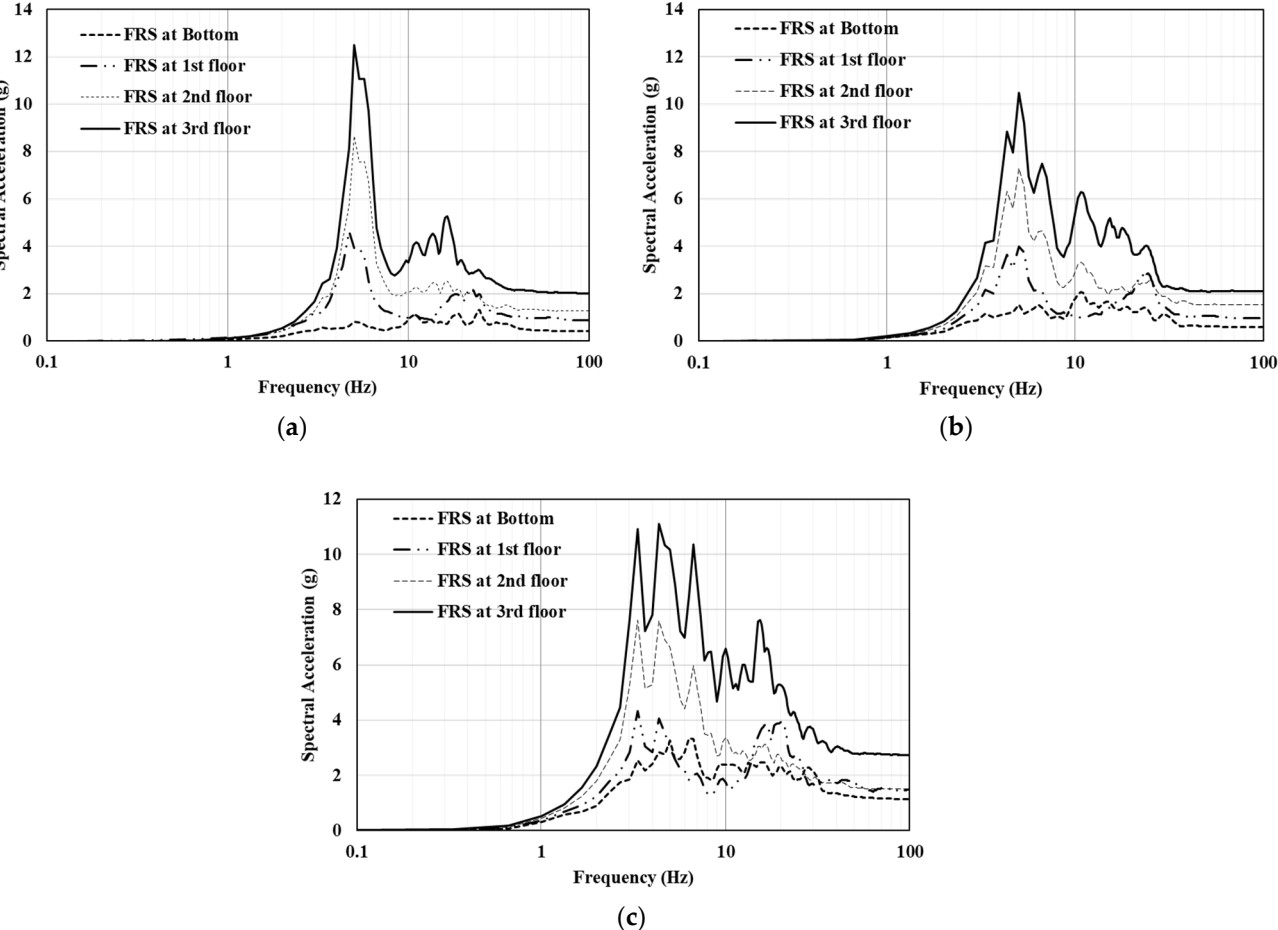

**Figure 10.** Floor Response Spectrum (FRS) for EQ#13, #17 and #19: (**a**) EQ#13; (**b**) EQ#17; (**c**) EQ#19.

In this study, Zero Period Acceleration (ZPA), Maximum Spectral Acceleration at high frequency (MSA), and top displacement response of the electrical cabinet were investigated. The limit-state of ZPA is defined as 1.8 g based on a previous study [24,25]. MSA means the maximum spectral acceleration between 15 and 30 Hz, where the response of the electrical cabinet system is significantly amplified. Figure 11 shows the ZPA and corresponding amplification factor of the cabinet, where the amplification factor is the ratio of peak floor acceleration at the floor of interest to peak ground acceleration. As damage to the structure increases, the ZPA on each floor increases, and the corresponding amplification factor decreases. However, the difference in the ZPA is not significantly higher during EQ#13 and #17, and the meaningful difference occurred in EQ#19. The difference in the amplification factor on each floor is not significant, and the deviation of the factor is reduced as the damage to the structure increases. In the minor and moderate damage states, all measurement points in the cabinet mounted on the third floor showed a functional failure. Some points, such as A10, A12, and A13 in the cabinet mounted on the second floor, also showed the functional failure. The first floor cabinet case shows the failure only at A10. However, in the major damage state, the functional failure occurs in most of the measurement points except for A9, A8, and A11 at the first floor cabinet. In addition, the responses in the first and second floor cabinet cases were upside down. Point A10 is located on a vertical panel perpendicular to the x-direction without any bracing to resist moving in the x-direction; thus, the response at A10 is significantly larger than the other responses. Figure 12 shows the MSA and the corresponding amplification factor of the cabinet. The result shows the same trend as ZPA. The values of SA at A10 in the third floor cabinet case were 47.7, 49.7, and 69.4 g, corresponding to EQ#13, #17, and #19, respectively. Figure 13 shows the top displacement (A13) of the cabinet and

the corresponding amplification factor. In the minor and moderate damage states, the displacement and amplification factors increased with the increase in the installation level of the cabinet. In the case of EQ#19, the response of the first floor cabinet case was larger than that of the second floor cabinet.

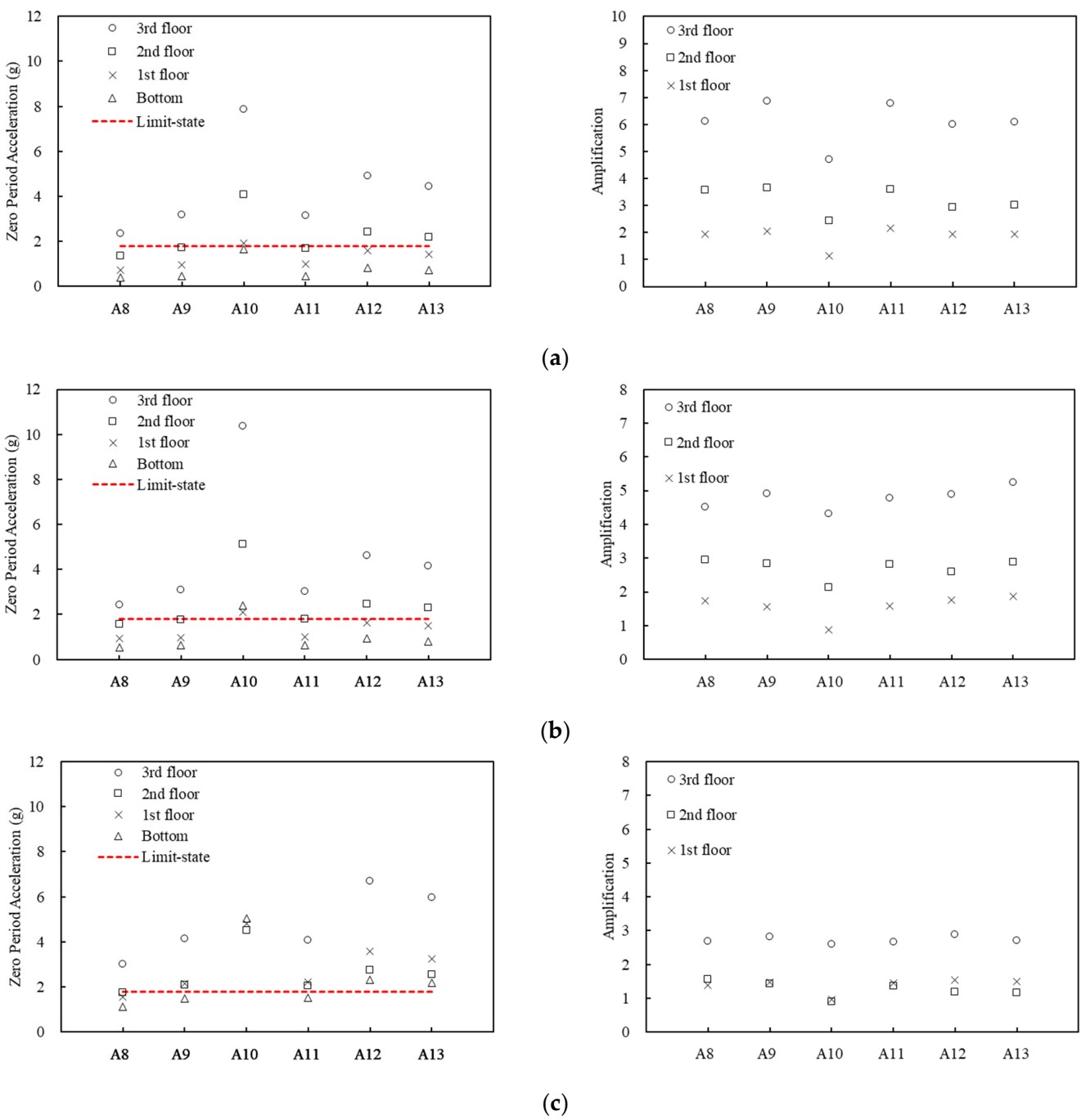

**Figure 11.** Zero Period Acceleration (ZPA) response and amplification factor: (**a**) EQ#13; (**b**) EQ#17; (**c**) EQ#19.

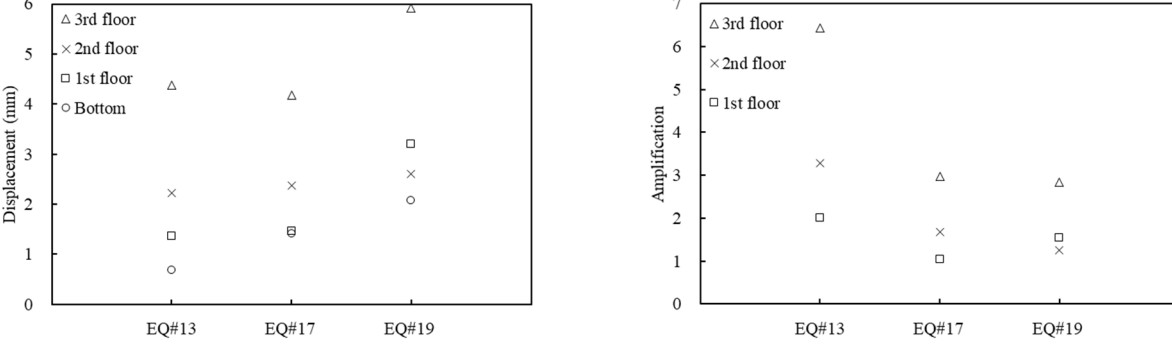

**Figure 12.** Maximum Spectral Acceleration at high frequency (MSA) and amplification factor: (**a**) EQ#13; (**b**) EQ#17; (**c**) EQ#19.

**Figure 13.** Top displacement response (A13) and amplification factor.

## 5. Conclusions

This study investigates the effects of the cabinet elevation and performance limit-state for a concrete shear wall structure on the response of a single-door electrical cabinet. For this purpose, a validated three-story concrete shear wall structure model and a single-door electrical cabinet model were used. Through a structural analysis of the concrete structure, the acceleration time responses on each floor were evaluated corresponding to a performance limit-state of the concrete structure. Based on the SMART-2013 experimental data and ACSE-43 standard, the levels of performance of the structure were defined as minor, moderate, and major damage states. Finally, a decoupled analysis of the electrical cabinet was conducted by applying the calculated floor responses on each floor to the cabinet model. Conclusions are as follows:

- With the increase in the damage to the structure, the cabinet's responses such as ZPA and MSA increased, but the corresponding amplification factor decreased. In addition, the cabinet's response was higher at higher elevations, with the exception of the cabinet responses subjected to the first and second floor motions of EQ#19. However, the increase or decrease in the response does not seem to be linear.
- The difference in the amplification factor at a floor is not significant, and the deviation of the factor is considerably reduced as the damage to the structure increases.
- In the minor and moderate damage states, all measurement points in the cabinet mounted on the third floor show a functional failure. Some points, such as A10, A12, and A13 in the cabinet mounted on the second floor, also showed the functional failure. The first floor cabinet case shows the failure only at A10. However, in the major damage state, the functional failure occurs in most of the measurement points except for A9, A8, and A11 at the first floor cabinet.
- There are significant differences in the cabinet's responses depending on measurement points in the cabinet. The measured responses at high positions (A10 and A9) tend to be greater than those at low positions (A8), and in particular, the A10 response shows a significantly larger response than the others. This is because A10 is located on a vertical panel perpendicular to the x-direction without any supports to resist movement in the x-direction. Thus, the seismic performance of the electrical equipment in the cabinet varies greatly depending on the location and mounting method of the electrical equipment in the cabinet.
- In the minor and moderate damage states, the difference in ZPA, MSA, and top displacement is not significant. However, a meaningful difference occurs in the degree of the major damage state. Thus, when the structure is under minor or moderate damage states, the responses in the cabinet are more sensitive to the installed floor level of the cabinet in the structure than to the intensity of seismic loadings.

**Author Contributions:** Conceptualization, B.-S.J., S.L. and S.K.; methodology, B.-S.J., S.L. and S.K.; validation, H.S. and S.L.; formal analysis, H.S. and S.L.; data curation, B.-S.J., H.S. and S.L.; writing—original draft preparation, B.-S.J., S.L. and S.K.;writing—review and editing, B.-S.J., S.L. and S.K. All authors have read and agreed to the published version of the manuscript.

**Funding:** This research was funded by the National Research Foundation of Korea (NRF) of the Korean Government (MSIT) grant number No.2021R1A2C1010278.

**Institutional Review Board Statement:** Not applicable.

**Informed Consent Statement:** Not applicable.

**Data Availability Statement:** The data presented in this study are available on request from the corresponding author.

**Acknowledgments:** This work was supported by the National Research Foundation of Korea (NRF) grant funded by the Korean Government (MSIT). (No.2021R1A2C1010278).

**Conflicts of Interest:** The authors declare no conflict of interest.

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
