# Peer review of "Estimating Seismic Demands of a Single-Door Electrical Cabinet System Based on the Performance Limit-State of Concrete Shear Wall Structures"

_sustainability, doi:10.3390/su14095480_

Round 1

Reviewer 1 Report

Please refer to the attachment (.pdf) for detailed comments.

Reviewer 2 Report

The paper deals with developing a 3D FEM model of shear wall structure subjected to several ground motions (mostly the scaled records of Northridge) corresponding to different levels of damages expressed by certain values of the inter-story drifts. Then, the acceleration response of each floor is determined and applied to the FEM model of an electrical cabinet through a decoupled analysis. The response of the cabinet is evaluated at some measurement points in terms of amplification factor, ZPA, MSA, and top displacement.

The topic is interesting to the researchers in the field, and it contains enough novelty for publication. The obtained numerical results are rational. The paper is well-written, and the English is satisfactory. Thereby, the manuscript is recommended for publication. A few minor points are listed in the following:

  • It is recommended to mention the reference for the software or computer code used for finite element analysis of the models.
  • Line 90: What is the basis for choosing 0.07g as PGA for sinusoidal shakes applied to the electrical cabinet?
  • Section 2.2: It is suggested to provide a brief note on the mechanical properties of materials used in the electrical cabinet.
  • Table 3: In the first row, the third column: EQ# needs to be corrected.
  • Line 202: Since the acceleration responses in three directions were determined (for point D of each floor), there is a question if these 3D acceleration records were applied to the electrical cabinet? Or instead, only one-directional acceleration is considered when analyzing the electrical cabinet? It is suggested to clarify this in the context.
  • Line 207: The caption for Figure 7 is repeated as a typo. So, please eliminate it.
  • References: Please add the following to the list of references:
  • Mahdavi G, Nasrollahzadeh K, Hariri-Ardebili MA. Optimal FRP jacket placement in RC frame structures towards a resilient seismic design. Sustainability. 2019 Jan;11(24):6985.

Round 2

Reviewer 1 Report

The corrections have been done as requested. This paper's quality has been improved. More discussion to support the data result may be added for further improvement (if possible).
